# Modeling Host–Pathogen Interactions in *C. elegans*: Lessons Learned from *Pseudomonas aeruginosa* Infection

**DOI:** 10.3390/ijms25137034

**Published:** 2024-06-27

**Authors:** Gábor Hajdú, Csenge Szathmári, Csaba Sőti

**Affiliations:** Department of Molecular Biology, Semmelweis University, 1094 Budapest, Hungary; szathmari.csenge@phd.semmelweis.hu

**Keywords:** innate immunity, virulence factors, pathogen recognition, cytoprotective stress responses, surveillance immunity, mitochondria, lysosome-related organelles, pathogen metabolite checkpoint, neuroendocrine regulation

## Abstract

Infections, such as that by the multiresistant opportunistic bacterial pathogen *Pseudomonas aeruginosa*, may pose a serious health risk, especially on vulnerable patient populations. The nematode *Caenorhabditis elegans* provides a simple organismal model to investigate both pathogenic mechanisms and the emerging role of innate immunity in host protection. Here, we review the virulence and infection strategies of *P. aeruginosa* and host defenses of *C. elegans*. We summarize the recognition mechanisms of patterns of pathogenesis, including novel pathogen-associated molecular patterns and surveillance immunity of translation, mitochondria, and lysosome-related organelles. We also review the regulation of antimicrobial and behavioral defenses by the worm’s neuroendocrine system. We focus on how discoveries in this rich field align with well-characterized evolutionary conserved protective pathways, as well as on potential crossovers to human pathogenesis and innate immune responses.

## 1. Introduction

The phenomenon of rapid host–pathogen coevolution is driven by the mutually advantageous long-term interplay of genetic variability and diversity through counter-adaptation mechanisms. In this intricate process, through a series of infections resulting in host disease, the host organism acquires functions associated with recognition of the invading species and the induced damage and mobilization of protective and restorative processes. In parallel, the parasitic organism develops more efficient colonization and mimicry strategies. This reciprocal pattern of antagonistic coevolution in nature is well documented and supported by experimental investigations, all aiming for a more comprehensive understanding of the underlying mechanisms [1,2,3].

A notable model organism to elucidate these dynamics is the nematode *Caenorhabditis elegans*. The worm gained popularity by its anatomical simplicity, conserved biological processes and molecular mechanisms, and well-characterized neuronal connectome, as well as its observable behavioral, learning, and memory patterns in response to environmental stimuli. Originally, these roundworms inhabit soil and decaying fruits, both of which harbor a multitude of bacterial species serving as a food source. Consequently, it is evident that several pathogenic bacteria recognize *C. elegans* as a suitable host organism, and this host–parasite interaction has been extensively studied in number of laboratories [4,5,6].

Importantly, worms lack dedicated, professional immune cells and adaptive immunity. Yet, as they inhabit rotten fruits and the soil and feeding with unicellular organisms, they developed highly efficient defenses, including an innate immune system and neural learning and memory to recognize and avoid pathogens [7,8,9]. They have a strong antimicrobial system to fight infection in the intestine, hypodermis, and nervous system, harboring both intracellular tissue defenses and conserved immune signaling pathways also present in mammals, including the p38 MAPK pathway, TGFβ signaling, regulation of gene expression by stress-sensitive bZIP-type transcription factors, and nuclear hormone receptors [10,11]. Furthermore, as recent years have brought us a deeper understanding of the critical role of innate immunity in priming and optimizing host fitness, *C. elegans* offers a valuable tool to study ancient mechanisms of recognition and protection without confounding cross-talks from the adaptive, evolutionary recently appeared immune mechanisms [8,12,13,14,15].

An intriguing and valuable subject in host–parasite interaction studies of nematodes is the infection by the Gram-negative human opportunistic pathogen *Pseudomonas aeruginosa* [16]. The clinical significance of this species encompasses nosocomial infections, including sepsis, pneumonia, and post-operative infections, particularly in burn patients, hospitalized individuals, and those with compromised immune systems [17]. Patients with hereditary cystic fibrosis (CF) or chronic obstructive pulmonary disease (COPD) face heightened vulnerability to chronic colonization by *P. aeruginosa*, leading to severe pulmonary damage and potential mortality [18]. Notably, among several clinically relevant strains, the PAO1 and PA14 strains are distinguished as laboratory reference strains to screen for novel therapeutics, of which the PA14 strain appeared to be hypervirulent to both vertebrates and invertebrates, including *C. elegans* [16,19,20,21]. Noteworthy, PA14 was the first pathogen that led to the foundation of *C. elegans* as an excellent model for infection and host defense [21,22]. This review aims to summarize our knowledge on virulence factors and infection strategies of *Pseudomonas aeruginosa*, as well as conserved antimicrobial defenses employed by *C. elegans* that efficiently respond to pathogen invasion and toxicity.

## 2. *Pseudomonas aeruginosa* Virulence Factors and Infection

Virulence factors intend to facilitate colonization and impair vital host defense and functions, promoting the spread of infection. Virulence factors of *P. aeruginosa* can be categorized into three main groups: bacterial surface structures, secreted factors, and cell–cell interaction molecules [23]. The biofilm formation of *P. aeruginosa* facilitates both protection against the host’s immune system and communication between cells—quorum sensing—required for efficient colonization and chronic infection [24].

Summarized in Table 1, the virulence factors of *P. aeruginosa* are listed, indicating their pathogenicity in *C. elegans* and mammals.

When infecting mammals, *P. aeruginosa* strains employ a strategy involving exotoxin effectors in combination to expedite invasion [68,69,70,71,72,73]. These effectors include (1) Exotoxin U (ExoU), a phospholipase A2; (2) Exotoxin S (ExoS) and Exotoxin T (ExoT), which function as dual Rho-GAP and ADP-ribosylase effectors; (3) Exotoxin Y (ExoY), an adenylate cyclase with a yet unknown function in pathogenicity; and (4) Exotoxin A (ExoA or ETA), a similar ADP-ribosylating factor. ExoS and ExoA have been confirmed to actively contribute to pathogenicity, as strains with single loss-of-function mutations in these toxins exhibit reduced cytotoxicity. Both the PAO1 and PA14 laboratory reference strains produce ExoA to accelerate acute infection [19]. This toxin consists of a single polypeptide chain comprising three functional domains responsible for receptor binding, endosomal internalization, and ADP-ribosylation, respectively [74]. The ADP-ribosylation domain necessitates furin-mediated proteolytic cleavage within endosomal compartments for attenuation of translation via the inhibition of eEF2, akin to the mechanism observed in diphtheria-mediated effects [71].

Scientists have identified at least three distinct strategies used by the *P. aeruginosa* PA14 strain to infect and harm *C. elegans*, which can be separately studied in worms, depending on the specific cultivation conditions [16,22]. The so-called “fast killing” condition involves the massive secretion of toxic effectors and host toxicity, “slow killing” is characterized by intestinal infection and colonization, and “liquid killing” is mainly caused by compromised mitochondrial function by pyoverdines and phenazines [16]. Other main virulence factors of *P. aeruginosa* and, therefore, anti-infective chemical libraries can be studied via “liquid killing” assays [16,75]. ExoA is one of the widely characteristic virulence factors of PA14, which identically targets translational machinery of both mammalian and worm host cells. Furthermore, PA14 secretes pyoverdine siderophore to sequestrate iron as a high-affinity chelator causing a hypoxia-like state [76,77], phenazine-1-carboxylic acid, along with pyoverdine to compromise the mitochondrial UPR (UPR^MT^) [61,76,78], and generally produces phenazine-type toxins to alter redox homeostasis by disruption of the electron transport chain and OXPHOS [59,78,79].

What makes PA14 infection of nematodes special is its “Trojan horse” approach [80]. Initially, *C. elegans* is strongly attracted to the PA14 bacterial lawn, but this preference gradually turns into avoidance after prolonged exposure or upon a second encounter with the pathogen [81]. This unique behavior makes PA14 an efficient parasite and nematodes an ideal model for studying pathogen-related aversive learning and memory formation.

## 3. Routes of Pathogen Recognition

Across countless generations of evolutionary coexistence, the hosts’ innate immune systems *Caenorhabditis elegans* have developed intricate cell-autonomous and non-autonomous mechanisms to detect and confront infections [10,11,82]. These molecular processes encompass pathogen-specific and general responses: (1) microbe- or pathogen-associated molecular pattern (MAMP or PAMP) involving the detection of specific, conserved microbial motifs; (2) damage-associated molecular pattern (DAMP) recognition of endogenous danger signals released by damaged or dying host cells, both recognized by pattern recognition receptors (PRRs) such as Toll-like receptors, C-type lectins, and others; (3) surveillance immunity, sensing of the disruption of essential cellular processes by microbial toxins; and (4) pathogen metabolite checkpoints, a direct sensing of pathogen-associated metabolite(s), which has recently been identified in *C. elegans* [2,9,12,83,84,85,86]. In nematodes, the recognition of infection also involves a two-way, direct neuroendocrine communication between the nervous and the innate immune systems to modulate immunity, innate, and learned behavioral defenses [87].

In mammalian models, the TLR4 and TLR5 Toll-like receptor-dependent activation of caspase-1 by inflammasomes is essential for the recognition of MAMPs, subsequently triggering multiple interleukin signals. Early infection also induces the recruitment of polymorphonuclear neutrophils, leading to essential damage pattern (DAMP) signals by the emergence of serine proteases induced by tissue inflammation. Additionally, the *P. aeruginosa* T3SS effector system induces a series of responses in antigen-presenting cells, leading to further expanding the signals. Ultimately, the recognition of distinct patterns results in organism-wide responses, mainly by professional immune cells [9,17,69,70,73]. See Figure 1 for a simplified model of the mammalian immune response against *P. aeruginosa*. *C. elegans* evolutionarily lost the genetic set of professional pattern recognition. However, via partially overlapping transcriptional programs, worms are able to discriminate between several pathogenic bacteria [88]. Accordingly, there are several nematode orthologs of mammalian MAMP recognition and inflammation elements that were shown to be involved in specific PA14 responses, detailed further below [82]. All the above suggests the existence of a similarly sophisticated, yet unexplored, mechanism(s) of pathogen detection, from which the *P. aeruginosa* model of host–pathogen interaction offers valuable insights.

## 4. Cellular Stress and Immune Responses

Major regulators of stress responses in *C. elegans*, including the p38 MAPK, DAF-16/FOXO, SKN-1/Nrf2, and HSF-1, are also key players of pathogen-induced DAMP recognition and immune responses. As antimicrobial defenses destroy not only the pathogen but also impair host tissues and imposes a stress on the homeostasis, the coordinated regulation of the immune defenses and cytoprotective measures by these master regulators confers protection and facilitates regeneration. Not surprisingly, several aspects of these strongly conserved pathways show features in mammalian innate immunity and organismal protection. Please see Figure 2 for a summary of the major innate immune responses of *C. elegans* and Table 2 for their key genes and activating signals.

**Figure 2 ijms-25-07034-f002:**
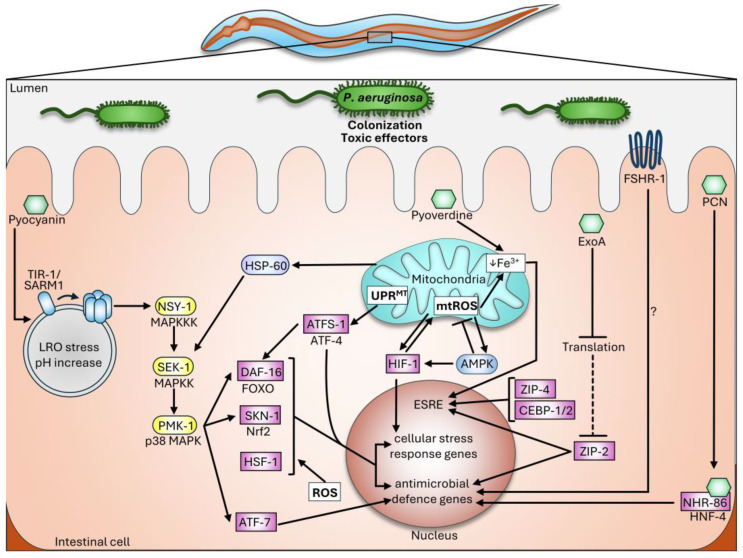
The innate immune response against *Pseudomonas aeruginosa* in the *C. elegans* intestine. See the text for details.

**Table 2 ijms-25-07034-t002:** Key *C. elegans* regulatory genes responding to *Pseudomonas aeruginosa* infection.

Key Regulators (Pathways)	Activatory Signals	Specific to *P. aeruginosa*	Relevance in Mammals	Ref.
TIR-1/SARM1 PMK-1/p38 MAPK (p38 MAPK pathway)	ROS, pyocyanin-induced LRO stress	No	Yes	[15,89,90]
DAF-16/FOXO (ILS and oxidative stress response)	ROS, proteostasis disturbance, suppressed DAF-2 signaling	No	Yes	[15,67,91]
SKN-1/Nrf-2 (Oxidative and electrophilic stress response)	ROS, PMK-1	No	Yes	[15,92,93,94,95]
HSF-1 (Heat shock response)	Proteostasis disturbance, ROS	No	Yes	[96]
FSHR-1	n.d.	No	No	[97,98]
ZIP-2 (Immune response, ESRE)	ExoA-induced suppression of mRNA translation, pyoverdine-induced mitochondrial dysfunction	Yes	No	[99,100,101,102]
ATFS-1/ATF4 (UPR^MT^)	Disturbance of mitochondrial proteostasis	n.d.	Yes	[103,104]
HIF-1, AMPK	Mitochondrial dysfunction due to pyoverdine-induced iron depletion	No	Yes	[56,77,105,106]
NHR-86/HNF4	Phenazin-1-caroboxamide	n.d.	No	[61,107]

n.d., not determined.

### 4.1. The p38 MAPK Pathway

Experiments in mammalian U937 cultures and THP-1 human monocytes revealed the role of the p38 MAPK signaling pathway in concert with NF-κB activation in orchestrating the recruitment of neutrophils to the site of acute *Pseudomonas aeruginosa* infection through the expression of interleukin 8 [89,108,109]. In both alveolar macrophages (AMs) and THP-1 cultures, the p38 MAPK pathway is also required to resist hyperactivation of the inflammation response via TLR4-specific MUC-1 expression [109]. Additionally, TLR4 and TLR5 Toll-like receptor, as well as IL-1R expressed in airway and alveolar epithelial cells, play a pivotal role in NF-κB-dependent secretion of neutrophil chemokines and antimicrobial surfactant proteins.

The nematodes lost both Rel genes, including NF-κB; hence, it may be that other innate defense mechanisms took on their role in the elicitation of the antimicrobial response seen in infected worm intestinal epithelium [12,73,86]. In agreement with this, the *C. elegans* mitogen-activated protein kinase p38 MAPK ortholog PMK-1 regulates basal and pathogen-induced expression of various antimicrobial proteins in *C. elegans*, such as C-type lectins, ShK toxins, and CUB-like effectors to combat infections [10,110]. Systematic analyses using loss-of-function mutants suggest that this evolutionarily ancient innate immune pathway may represent the primary and critical line of defense against PA14. Interestingly, when exposed to PA14, only a quarter of PMK-1-dependent genes were induced, suggesting a much broader spectrum of microbial recognition with the potential specificity of the respective pathogens [110,111]. This is further supported by the heightened susceptibility of *pmk-1* loss-of-function mutant strains exposed to several Gram-positive and Gram-negative pathogens [110,112]. The Toll/IL-1 receptor domain protein TIR-1/SARM and the upstream NSY-1–SEK-1 MAPK kinase cascade leading to PMK-1 activation has been elucidated through *P. aeruginosa* infection studies [112]. An interesting discovery revealed the involvement of the p38 MAPK pathway from worms to mammals in PA14 infection resistance induced by the antidiabetic drug metformin, indicating an evolutionary ancient and effective way of immune priming [113]. An intricate cell non-autonomous role of the intestinal p38 MAPK pathway was demonstrated as the PMK-1 target. ATF-7 transcription factor was responsible for protection against neuronal loss of mitochondrial function [114]. PMK-1 activity and the production of immune peptides burdening the ER requires a protective upregulation of the UPR^ER^ by IRE-1/XBP-1 route [115].

### 4.2. TIR-1/SARM Activation and Immune Surveillance of Lysosome-Related Organelles

Recent findings have connected antimicrobial defenses and pathogen recognition by the PMK-1 pathway to subcellular vesicles, so-called gut granules. Gut granules are primordial lysosome-related organelles (LRO) in the *C. elegans* intestine [116]. First, it was shown that PA14 infection results in the deposition of autofluorescent material in gut granules and the induction of the *glo*-1 and *pgp-2* LRO biogenetic genes, which are required for pathogen resistance against PA14 [117]. Then, a current preprint from the Pukkila-Worley group demonstrated that the PA14-secreted phenazine pyocyanin-induced oxidative stress alkalinized and collapsed LROs, which led to the aggregation and enzymatic activation of the LRO-associated TIR-1/SARM, providing the missing mechanistic link between pathogen infection and PMK-1 p38 pathway activation [90].

Importantly, cholesterol deficiency also activates TIR-1 [118], and benzaldehyde exposure induces toxicity, as well as transcriptional and functional LRO expansion [117,119], which both confer increased resistance against PA14 infection and other abiotic stresses. Such findings are consistent with the role of nematode LROs in stress sensing and thus in surveillance immunity. Based on the severe deficiency of innate immune responses in human genetic LRO disorders [120], an orthologous function might be plausible, although it awaits proof.

### 4.3. The DAF-16/FOXO, SKN-1/Nrf2, and HSF-1 Pathways

PMK-1 function is indispensable of the *C. elegans* insulin-like signaling pathway and SKN-1/Nrf-2-regulated defenses. The increased resistance to pathogens observed in the insulin-like receptor ortholog *daf-2* loss-of-function animals, mediated by hyperactivated FOXO/DAF-16, depends on functional PMK-1 [110,121]. Furthermore, PMK-1 is necessary for SKN-1 activation in response to reactive oxygen species (ROS) and PA14 infection [92,122]. SKN-1 is required for resistance against PA14 and other pathogens, mediates the immune-boosting effects of ILS mutants and hormetic oxidative preconditioning, and its functional decline is responsible for immunosenescence [92]. Since the above processes are not exclusive to *P. aeruginosa* infection, DAF-16 and SKN-1-regulated transcription in nematodes contribute to immunity as a rather general response against oxidative and electrophilic stress resulting from pathogen invasion and pathogen-produced metabolites. Not surprisingly, mammalian FOXO and Nrf2 are required in alveolar and bronchial epithelial cells for fine-tuned innate immune responses [91,123]. However, despite direct PMK-1 phosphorylation sites on both DAF-16 and SKN-1, there is only little or no overlap between PMK-1 and DAF-16, as well as PMK-1 and SKN-1, regulated immune-related gene expressions [110,124,125,126], indicating different subsets of target effectors. A functional p38 MAPK pathway is also crucial for the extended longevity observed in *daf-2* mutants and for various age-related genes regulated by SKN-1 [92,121,122]. Additionally, insulin-like signaling plays a vital role in neuroendocrine fine-tuning of innate immunity in the intestine orchestrated by PMK-1 [127].

These conserved stress pathways are also linked to the *C. elegans* heat shock transcription factor 1 (HSF-1) regulated heat shock response. HSF-1 detects misfolded, denatured proteins in response to proteotoxic insults, such as heat shock and oxidative stress during infection [128]. Besides molecular chaperones, HSF-1 drives the expression of metabolic and immune effector genes, including DAF-16 targets [128,129]. Consistent with this, *hsf-1* mutants not only exhibit defective microbial, including PA14, resistance, but they are short-lived and die due to the intestinal proliferation of the opportunistic *E. coli* OP50 [130]. Likewise, HSF-1 and DAF-16 collaborate to protect proteostasis during PA14 infection [131]. HSF-1 also regulates DAF-16 nuclear export [96,125,130]. Upon heat shock of *daf-16* overexpressed or *hsf-1* RNAi strains, excessive presence of DAF-16 in the nuclei has been demonstrated to be deleterious against a subsequent pathogen challenge [125]. More recently, olfactory stimulus by *P. aeruginosa*-related cues has been proven to enhance HSF-1-mediated expression of chaperones, as well as the aversion of PA14 lawn upon a secondary infection, showing a neuroendocrine priming of HSF-1-dependent antimicrobial defenses [132]. Altogether, these results indicate that general stress response pathways governed by DAF-16, SKN-1, and HSF-1 actively participate in the physiological response to infection.

### 4.4. The FSHR-1 Pathway

The G-protein-coupled mammalian follicle stimulating hormone receptor homolog FSHR-1 contains a highly conserved leucine-rich repeat (LRR) domain across species [133]. In *C. elegans*, FSHR-1 is primarily found in the intestine, where, in parallel with the p38 MAPK pathway, it plays a crucial role in the activation of innate immune effectors [97]. Its function is to detect and combat colonization by both Gram-negative and Gram-positive pathogens, providing a general antipathogen monitoring system. However, how FSHR-1 detects infection is currently unknown. Notably, studies have shown that disrupting FSHR-1 through RNAi treatment or loss-of-function mutations significantly impacts survival when confronted with various pathogenic bacteria, including *P. aeruginosa*. Strikingly, FSHR-1-dependent protection includes detoxification against heavy metal and oxidative stresses as a plausible crosslink between this pathway and SKN-1 orchestrated processes [134]. This response was demonstrated to be cell-autonomous, as the intestine-specific expression of *fshr-1* successfully rescued pathogen resistance [97]. On the other hand, the FSHR-1 cell non-autonomously regulates the activation of the mitochondrial unfolded protein response, which might be a key element of organismal mitohormesis and ROS surveillance, further discussed below [135]. At present, FSHR-1-depedent defenses appear nematode-specific, as a similar role for mammalian FSHR has not been shown [98].

All of the above indicate that, although *C. elegans* lacks an innate immune signaling axis of interferons, such as RIG-I-like receptor and NF-κB, as well as professional antigen presentation, fine-tuned intracellular and inter-tissue signaling of pathogen recognition and defenses are highly conserved.

## 5. Surveillance Immunity of ExoA-Mediated Translation Inhibition

A distinctive aspect of *Pseudomonas aeruginosa* infection of *C. elegans* is its ability to mimic harmless, high-quality food bacteria through the secretion of compounds to alter the recognition of danger, thereby triggering attraction and downregulation of cytoprotection during the initial encounter [81,136,137]. In response to nutrition, the neuroendocrine system of worms suppresses intestinal immune protection via secreted insulin-like ligands, as well as aversion via intestinal inhibition of learning and cell non-autonomous induction of detoxification [137,138,139,140,141,142]. By the time nematodes initiate cytoprotection and aversion by modulating neuroendocrine signaling in response to the pathogenic threat, PA14 has already colonized the intestinal lumen, starting to disrupt core cellular processes, such as translation and mitochondrial function. The foremost recognized sign of infection is provided by ExoA toxicity, which shares characteristic phenomena in mammals and nematodes. More precisely, it disrupts *C. elegans* translational machinery in intestinal epithelial cells by cleavage of the ribosomal decoding center [99,143]. Experiments on *C. elegans* IRG-1 (Immune Response Gene 1) was a milestone suggesting the existence of a distinct immune pathway, as its expression was induced specifically by *P. aeruginosa*, independently of the canonical p38 MAPK and FSHR-1 pathways [101]. The screen using a fluorescently labeled construct of IRG-1 led to the discovery of the ZIP-2 pathway, considered as a secondary layer of defense as bZIP-type transcription factors appearing lately in evolution. Interestingly, ZIP-2 regulation of the immune response was exclusively linked to *P. aeruginosa* as of yet, also providing evidence for surveillance immunity in *C. elegans* [99].

Mechanistically, translation inhibition by ExoA induces the expression of ZIP-2 via translational attenuation in a conserved uORF region [99,143]. Interestingly, a distinct set of ZIP-2 target genes were demonstrated upon PA14 or ExoA producing OP50 *E. coli*, confirming a complex pattern recognition of the damage. The role of endocytosis was also demonstrated, as the transcriptional induction of all tested immune effectors were reduced when endocytosis was abrogated [99]. In mammalian cell cultures, abrogated endocytosis by methylamine treatment was demonstrated to prevent ExoA toxicity [72,144]. The response to pathogen colonization by the *C. elegans* ZIP-2/*irg-1* axis was demonstrated to be an early response compared to the canonical p38 MAPK pathway or FSHR-1 activation; however, proper activation of all three pathways was necessary to overcome ExoA toxicity [35]. In human whole blood samples, ExoA was demonstrated to specifically bind to the alpha2-macroglobulin receptor, which interaction induces a specific, dose-dependent, and tumor necrosis factor (TNF)-independent repression of antimicrobial cytokine production and monocyte activating factor CD14 [72]. As the current knowledge shows, there is no mammalian ortholog of ZIP-2; instead, exotoxin A is an essential inducer of cytokine storms in macrophages and T cells [102].

The ZIP-2 transcriptional pathway’s specificity enabled the coevolution of a precise neuronal upregulation of swift behavioral avoidance and ZIP-2-dependent antimicrobial response triggered by the AWB chemosensory recognition of PA14-secreted 1-undecene [145]. This suggests an evolutionary development of avoidance learning over generations of infections, as a single odor compound was discriminated and associated with the disruption of an essential process of translation.

## 6. Surveillance Immunity of Mitochondrial Function

The proper functioning of mitochondria within epithelial cells of *Caenorhabditis elegans* is essential for the maintenance of metabolic, energy, and redox homeostasis. The surveillance of mitochondrial function requires the coordinated activity of various pathways to detect unfolded proteins, reactive oxygen species (ROS), hypoxia, and low ATP concentration [146,147,148]. Consequently, the ability to either disrupt or protect mitochondrial function is a critical aspect of the antagonistic interplay between *P. aeruginosa* and *C. elegans*. This dynamic interaction involves intricate mechanisms related to both the previously detailed pathogen-secreted effectors and host defense [103,146,147]. The recognition and restoration of the pathogen-associated disruption of mitochondrial function is almost exclusively investigated in response to PA14, including both evolutionary ancient and alternative pathways of surveillance immunity.

### 6.1. Stress Response Element (ESRE)

One notable pathway involves a cluster of stress-induced genes containing the Ethanol and Stress Response Element (ESRE) motif [100]. This network is highly conserved and detects mitochondrial damage induced by a wide range of abiotic stresses. Notably, it has been demonstrated that the ESRE network is upregulated in response to damage caused by *P. aeruginosa* pyoverdines [77]. This upregulation is dependent on ZIP-2 transcriptional regulation and other bZIP transcription factors. As far as the current knowledge shows, the ESRE network primarily functions to monitor mitochondrial homeostasis but does not directly mediate antimicrobial effects.

### 6.2. Mitochondrial Unfolded Protein Response (UPR^MT^)

Mitohormesis, the surveillance of mitochondrial functions, is a convergent defense mechanism against not only abiotic stresses but also various pathogenic assaults [148]. Importantly, a mild alteration in mitochondrial function strengthened stress and immune protection via the PMK-1/p38 MAPK and DAF-16/FOXO-mediated routes [149]. Within this context, the ATF4 ortholog ATFS-1 plays a pivotal role by orchestrating the UPR^MT^ in response to proteotoxic stress [103,147,149,150]. ATFS-1, a bZIP-type transcription factor, translocates to the nucleus when its nuclear localization signal (NLS) becomes exposed upon the accumulation of dysfunctional and aggregated mitochondrial proteins. Subsequently, it triggers the expression of mitochondrial chaperones in the nucleus. In response to PA14 infection, the nuclear translocation and activation of ATFS-1 drives the expression of antimicrobial peptides, including lysozymes and C-type lectins [147]. Furthermore, exposure to PA14 not only leads to the upregulation of conventional mitochondrial chaperones *hsp-6* and *hsp-60* in an *atfs-1*-dependent manner but also triggers the expression of the ZIP-2 target *irg-1* [103]. It is noteworthy that there is no apparent induction of UPR^MT^ in response to the *P. aeruginosa* ΔgacA strain, which is defective in the synthesis and secretion of ExoA. Additionally, *zip-2* expression exhibits some dependence on *atfs-1*, implying a connection between mitochondrial stress and the response elicited by ExoA [103]. The UPR^MT^ is intricately linked to detoxification through the HSP-6-mediated expression of cytochrome P450 CYP-14A, a process regulated by the nuclear hormone receptor NHR-45 [151]. Furthermore, UPR^MT^ is also interconnected with the canonical p38 MAPK pathway via a unique cytosolic interaction between HSP-60 and MAPKK SEK-1 in both intestinal cells and neurons [152].

### 6.3. Siderophore-Specific Mitophagy

The selective removal of impaired mitochondria through macroautophagy is a fundamental cellular mechanism of energy and nutritional maintenance [153]. Notably, disruption of the mitochondrial iron balance by *P. aeruginosa*’s pyoverdine triggers not only the Ethanol and Stress Response Element (ESRE) network but also a more general mitophagic response aimed at recycling damaged organelles [77,154]. It is worth highlighting that mitophagy is specifically responsive to PA14 liquid intoxication and exhibits similar effects in response to abiotic iron extraction [77,154] and that mitochondria-targeting pyoverdines exhibit a similar pattern of accumulation to that of the *C. elegans* intestinal LROs, perhaps as an attempt at sequestration [77,117]. This suggests that the clearance response primarily focuses on alterations in the overall mitochondrial homeostasis, in addition to promoting antimicrobial activity. Supporting this notion is the observation that siderophore-specific pathogen resistance is ameliorated through the canonical p38 MAPK pathway but is contingent upon the ZIP-2 axis of the immune response [77]. An explanation for this phenomenon could be that the activation of the p38 MAPK pathway inadequately shifts resources upon pyoverdine toxicity, whereas the ZIP-2-mediated antimicrobial defense serves to address the overall disruptions in cellular functions related to proteostasis and energy balance.

### 6.4. Mitochondrial Reactive Oxygen Species (mtROS)

The regulation of mitochondrial reactive oxygen species (mtROS), the harmful byproducts of the electron transport chain, is crucial for maintaining metabolic balance, redox homeostasis, and longevity [148,155]. Mitohormesis refers to the constant monitoring and potential activation of evolutionary mechanisms such as HIF1α-governed transcription and non-transcriptional signaling pathways such as mitophagy, tissue-specific NADPH oxidase (NOX), and superoxide dismutase (SOD) [106,156,157]. It is also crucial to limit harmful ROS levels through Nrf2-regulated antioxidant and oxidative stress response pathways via peroxiredoxin, thioredoxin, and glutathione-associated enzymes [158,159].

In mammals, the dysregulated generation of reactive oxygen species (ROS) results in persistent oxidative stress, predisposing individuals to pathological conditions such as cardiovascular or neurodegenerative diseases [156]. In mouse macrophages and hypoxic liver, activation of the TLR4 and TLR5 receptors triggers NF-κB, leading to enhanced HIF-1α transcription [73]. In *C. elegans*, an essential aspect of this regulation involves the interaction between the energy sensor AMPK and the nematode HIF-1, which together balance mtROS production [105]. Activation of the *C. elegans* SKN-1-dependent detoxification machinery through the p38 MAPK pathway has been observed in response to increased cytosolic levels of reactive oxygen species (ROS) [160].

When mitochondrial disruption occurs due to pathogen invasion, both in mammals and nematodes, HIF-1 adjusts mtROS production by increasing the availability of free iron [106,161]. This response is particularly important, since the PA14 toxin pyoverdine depletes iron. Moreover, elevated PA14 resistance by H_2_O_2_ preconditioning requires both the SKN-1 and DAF-16 transcription factors [92]. Similar mechanisms might regulate self-protection when mtROS is leaking or HSP-60 translocates to the cytosol [152]. Simultaneously, a negative feedback loop controls mtROS production through direct HIF-1 regulation by AMPK, mitigating the toxic effects of mitochondrial free radicals [78,105].

## 7. Alternative Pattern Recognition by the Pathogen Metabolite Checkpoint

The concept of a “pathogen metabolite checkpoint” in response to *P. aeruginosa* infection may represent an evolutionary countermeasure to address the pathogenic bacteria’s “Trojan horse” strategy [84]. The ability of PA14 to dampen antimicrobial responses (such as IL-1 in mammals and INS-7 and ZIP-3 in *C. elegans* [67,68,162]) and avoidance behaviors (like INS-11 [142]) might have contributed to the development of a similarly sophisticated response. This intricate way of recognition can be defined as a direct detection of, and innate immune response by the host immune system to a secondary metabolite produced by the pathogen. Worms possess a 284-membered nuclear hormone receptor family (NHR), which might be the core of a yet undiscovered large network of homeostasis surveillance systems governed by a specific ligand (i.e., pathogen-secreted toxin) recognition. A recent discovery has unveiled the role of a nuclear hormone receptor, NHR-86, in orchestrating an intestinal antipathogen transcriptional program [61,107]. This program includes the activation of PMK-1 target effectors in response to the secondary metabolites, PCN and PCH, which was found to be independent of the DAF-7-elicited aversion [61]. Remarkably, it has been demonstrated that the immunostimulatory effect of both the artificial inducer R24, as well as synthetic PCN, operates through NHR-86. This was highlighted by the observation that animals with a degraded NHR-86 protein experienced severe toxicity when exposed to PCN, which is otherwise only mildly toxic [107]. It is noteworthy that, among others, the expression of the NHR-86 target gene *irg-5* was upregulated in response to both PA14 and R24 and conferred protection against PA14 infection [107]. Moreover, the induction of the NHR-86 target *irg-5* and other antimicrobial effectors required the *glo-1* and *pgp-2* LRO biogenesis genes [117].

To gain a deeper understanding of how pathogen metabolites elicit an immune response, a cell non-autonomous increase in the transcription of antimicrobial peptides regulated by NHR-49 was discovered [163]. Interestingly, this set of NHR-49 target genes differs from those upregulated in germline-less *glp-1* mutants, which promotes longevity in a DAF-16-dependent manner. Furthermore, the rescue of NHR-49 in neurons restores resistance to PA14, while its hyperactivation in both neuronal and intestinal cells enhances survival [163], indicating a non-neuronal priming as well. Importantly, NHR-49 gained an independent hit of a PMK-1/p38 MAPK-dependent antimicrobial function via a compound screen against *P. aeruginosa* pathogenesis, while NHR-45, responsible for monitoring mitochondrial homeostasis and the defense against *P. aeruginosa*-secreted toxins, was not implicated in the same screen [75,151]. Another member of the immune homeostasis responsive nuclear hormone receptor family, NHR-14, has been efficiently adapted to regulate the transcription factors’ PQM-1 and DAF-16-dependent innate immune gene expression in the intestines by monitoring iron homeostasis [164], probably by the direct binding of Fe^2+^ in peripheral tissues.

In summary, toxic *P. aeruginosa* secondary metabolites trigger immediate and extensive antimicrobial responses by specific metabolite recognition of NHR family members and perhaps other receptors. As detailed above, the “pathogen metabolite checkpoint” delicately links pattern recognition to surveillance immunity (i.e., disruption of *P. aeruginosa*-targeted core cellular processes) by inducing a broad spectrum of stress and innate immune pathways. Several pieces of evidence above demonstrate a possible neuro-intestinal link of these responses to arm efficiently fine-tuned weapon against the multifaceted PA14 challenge.

## 8. Neuroendocrine Regulation of Physiological and Behavioral Defenses

The ability to distinguish between safe and hazardous microorganisms is a fundamental aspect of organismal fitness [3,9]. This requires the continuous and precise monitoring of external (chemosensory) and internal (physiological homeostatic) signals by the host’s nervous system and the rapid initiation of effective defense responses against danger [85]. This is especially true for *C. elegans* living in a niche enriched in microorganisms, both food sources and pathogens. *C. elegans* detects environmental cues, such as the presence of a new food source, through its sensory neurons. This detection leads to distinct behavioral, cytoprotective, and immunological outcomes: when bacteria are perceived as harmless, the worm approaches and feeds on them; otherwise, armed with the evolutionarily ancient “fight-or-flight” response, sensory neurons of the worm elicit behavioral aversion (flight) and/or an innate immune response (fight) against dangerous bacteria [6,11]. However, the invasion of *P. aeruginosa* challenges the “fight-or-flight” response in *C. elegans,* because the pathogen disrupts the nematodes’ ability to recognize danger in time [67,162,165]. Yet, not only pathogen-derived chemosensory cues but metabolites and the host’s internal physiological signals, may all activate the “fight-or-flight” response via the neuroendocrine system [136]. Various protective neuroendocrine mechanisms of *C. elegans* are summarized in Figure 3.

A single pathogen-derived odorant, 1-undecene, innately triggers protective responses mediated by an AWB chemosensory neuron pair [145]. In a similar scenario, the activation of immune effectors and aversive behavior governed by ASJ neurons in response to PCN, pyochelin, and nitric oxide produced by PA14 were demonstrated [57]. Noteworthy, upon infection, TGF-β-dependent signaling of ASJ not only induces activation of adjacent interneurons, but this recognition response might be simultaneously counterbalanced by the pathogenic bacteria through inducing a suppressive neuropeptidergic signaling in both ASJ and AWB via NPR-8, as one key element of the “Trojan horse” mechanism [57,165]. Although there is no exact information on which (invading or protecting) neuronal signaling pathway comes first in the evolution, interestingly, preconditioning with PCN and PCH compounds leads to the expression of immune effectors and enhances resistance against subsequent PA14 infections, comparable to those induced by the immunostimulatory R24 compound [107].

It was found that *C. elegans* learns to avoid the PA14 lawn represented a hypoxic milieu via NPR-1 receptor signaling in the oxygen-sensing neurons [57,166]. Utilizing 1D and 2D NMR analyses, researchers identified that the altered aerotaxis behavior is influenced by the PA14 secondary metabolites phenazine-1-carboxamide (PCN) and pyochelin (PCH) but not by pyoverdine or exoA [57,138]. Another intricate way to sense and avoid a hypoxic milieu is the recognition of elevated CO_2_ concentrations by the sole Toll-like receptor ortholog TOL-1-mediated gene expression in the chemosensory BAG neurons, which was independent of the canonical TIR-1–SEK-1–NSY-1 p38 MAPK pathway [167]. NPR-1, TGF-β, and insulin signaling pathways are also involved in the fine-tuning of CO_2_-elicited behaviors [168].

The innate recognition of compounds produced by or related to PA14 represents an effective strategy, but it is crucial to note that only prolonged exposure to the pathogen lawn generates a sufficient quantity of these metabolites to trigger organismal responses. Fresh *P. aeruginosa* lawns, on the other hand, have the ability to suppress both “fight” and “flight” responses of the host. Firstly, *C. elegans* is attracted to fresh PA14 lawns in response to pyrroles sensed by AWA neurons [169], as well as to unknown compounds sensed by AWC neurons [170]. Secondly, once worms are already feeding on the pathogenic bacteria, quorum sensing regulators encoded by lasR and rhlR, in conjunction with the previously mentioned exoA, suppress the expression of DAF-16-dependent immune effectors. This suppression is achieved through the overexpression of the insulin-like peptide INS-7 upstream of the DAF-2 receptor [67]. Thirdly, PA14 also suppresses pathogen recognition and the antimicrobial defense by modulating classical neurotransmitter pathways: (i) the neuropeptides NPR-1 (detailed below) and NPR-8 inhibit aerotaxis and cuticular collagen synthesis, respectively [138,165,171]; (ii) octopamine, produced by the RIC interneuron, inhibits immunity by binding to the OCTR-1 GPCR expressed by ASH neurons [172]; (iii) dopamine released by CEP neurons binds the DOP-4 receptors on ASG neurons and efficiently suppresses immune effector expression and PA14 resistance [173]; and (iv) the TRPV-channel OSM-9/OCR-2 dampens AMPA-type glutamaterg neurotransmission [174].

Prolonged exposure to *P. aeruginosa* leads to an enhancement in the host’s innate immunity (“fight”) and/or behavioral avoidance (“flight”) [81,87,136]. These responses can be induced in parallel or switched between, depending on chemosensory cues and the extent of physiological damage [83,175]. For instance, the neuropeptide Y receptor NPR-1, expressed in URX, PQR, and AQR oxygen-sensing neurons, amplifies both the aversion and innate immunity in response to a PA14 lawn when the oxygen concentration is low [138]. Conversely, during the early stages of infection, NPR-1-induced aversion is suppressed by the ubiquitin ligase HECW-1, expressed by the OLL labial neuron pair [176]. Furthermore, HSF-1-mediated olfactory priming of the transcriptional immune response required 5-HT signaling, which was also indispensable for olfactory stimulus-mediated early aversion of the PA14 lawn [96,132]. Simultaneously, the chemosensory ASER neuron-specific, as well as cholinergic motoneuronal SOD-1, facilitate efficient detoxification to combat pathogen invasion and prevent aversion [141,174,177]. As the infection progresses and the colonization of the intestine overwhelms the detoxification driven by SOD-1, behavioral avoidance begins to emerge. The role of extraneuronal (i.e., intestinal) damage in shaping behavior has been substantiated further by abiotic disruption, RNAi targeting of the core cellular processes, and by distension of the intestinal lumen due to prolonged *P. aeruginosa* infection by serotonergic and neuropeptidergic signaling, respectively [178,179]. It is also demonstrated that intestinal bloating induces microbial aversion in the next generation by germline histone acetylation [180]. Finally, it has been shown that mitochondrial damage and UPR^MT^-induced microbial aversion also employ neuroendocrine signaling. The RIC modulatory neuron responds to extraneuronal damage by releasing octopamine, which, in turn, activates aversive memory through the AIY interneuron [181]. In summary, bidirectional communication between intestinal and neuronal cells during infection serves as a selective mechanism to detect threats and fine tune responses accordingly.

## 9. Summary and Outlook

The measure of success in a host–parasite coevolution is the speed and efficiency of novel strategies of infection and defense, illustrated by the intricate race between *Caenorhabditis elegans* and *Pseudomonas aeruginosa*. The main routes of *P. aeruginosa* infection and recognition in mammalian cells have been thoroughly reviewed, although, as yet, far from understood and efficient therapeutic interventions are missing [182]. Besides the results summarized in this review, the *C. elegans* model may offer deeper mechanistic insights into *Pseudomonas* pathogenesis and toxicity. Further research may delineate the spatiotemporal organization of host defenses involving the paracrine and neuroendocrine coregulation of cell-autonomous stress responses, innate immunity, and the nervous system. A simple nervous system facilitates the investigation of neuroimmune communication and the coordination of behavioral and immune defenses. At the meta-organismal level, the worm is a promising model to study the role of microbiota in antimicrobial defense. Finally, it is a convenient platform for screening various compound libraries to find potential therapeutic interventions.

## Figures and Tables

**Figure 1 ijms-25-07034-f001:**
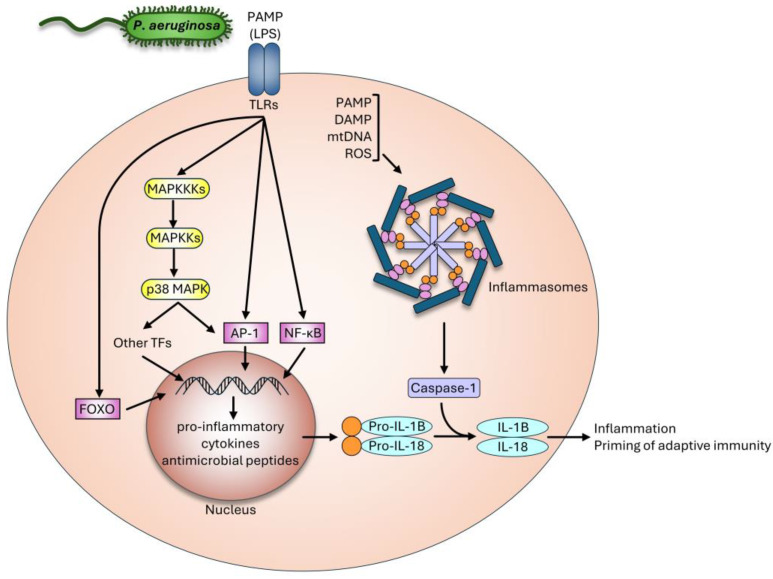
A simplified model of the mammalian innate immune response against *Pseudomonas aeruginosa*. Please see Figure 2 and Table 2 for shared and private pathways between mammals and *C. elegans*.

**Figure 3 ijms-25-07034-f003:**
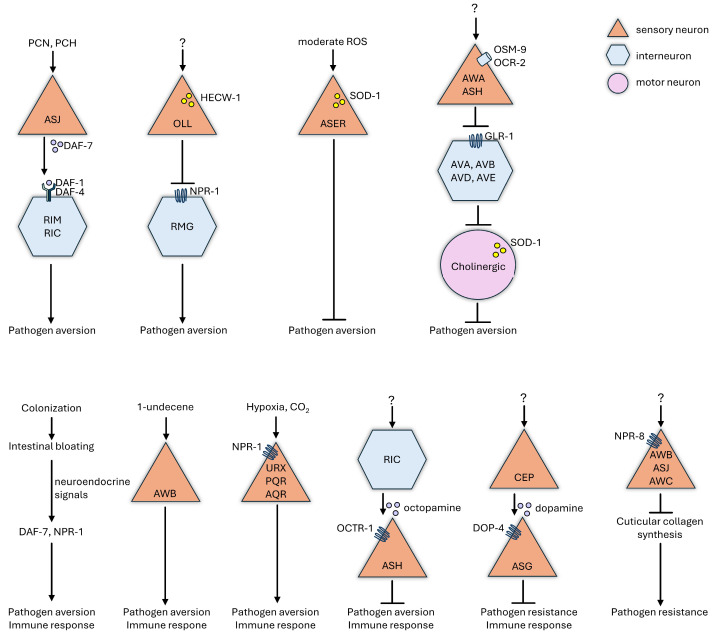
Neuroendocrine mechanisms protecting *C. elegans* against *Pseudomonas aeruginosa* infection. See the text for details.

**Table 1 ijms-25-07034-t001:** *Pseudomonas aeruginosa* virulence factors.

Virulence Factors	*C. elegans*	Mammals	Ref.
**Bacterial surface structure**			
Flagella (surface appendages)	+	+	[25,26]
Type IV pili (surface appendages)	+	+	[27,28]
Lipopolysaccharide (OMC)	+	+	[29,30]
**Secreted factors**			
Alkaline protease A (protease, T1SS)	−	+	[31,32]
Alkaline protease X (protease, T1SS)	n.d.	n.d.	[33]
HasAp (heme acquisition protein, T1SS)	n.d.	n.d.	[34]
ExoA (toxin, T2SS)	+	+	[35,36]
Lipase A (toxin, T2SS)	n.d.	n.d.	[37]
Lipase C (toxin, T2SS)	n.d.	n.d.	[38]
Phospholipase C (toxin, T2SS)	n.d.	+	[39]
Protease IV (protease, T2SS)	n.d.	+	[40]
Elastase A (protease, T2SS)	n.d.	+	[41]
Elastase B (protease, T2SS)	−	+	[31,41]
ExoS (toxin, T3SS)	n.d.	+	[42]
ExoT (toxin, T3SS)	+	+	[42,43]
ExoU (toxin, T3SS)	+	+	[42,43]
ExoY (toxin, T3SS)	+	+	[43,44]
PemA (effector, T3SS)	−	−	[43,45]
PemB (effector, T3SS)	+	−	[43,45]
EstA (esterase, T5SS)	n.d.	n.d.	[46]
TpsA (exoprotein, T5SS)	n.d.	n.d.	[47]
TpsB (β-barrel transport protein, T5SS)	n.d.	n.d.	[47]
LepA (protease, T5SS)	n.d.	+	[48]
Tse1 (bacteriolytic effector, T6SS)	n.d.	n.d.	[49]
Tse2 (bacteriolytic effector, T6SS)	n.d.	−	[50]
Tse3 (bacteriolytic effector, T6SS)	n.d.	n.d.	[49]
Alginate (exopolysaccharide)	−	+	[51,52]
Pel (exopolysaccharide)	+	−	[53,54]
Psl (exopolysaccharide)	+	−	[53,54]
Pyoverdine (siderophore)	+	+	[55,56]
Pyochelin (siderophore)	+	+	[55,57]
Lipoxygenase (toxin)	n.d.	+	[58]
Phenazine-1-carboxylic acid (toxin)	+	+	[59,60]
Phenazine-1-carboxamide (toxin)	+	n.d	[61]
1-hydroxyphenazine	+	+	[59,62]
Pyocyanin	+	+	[21,63]
**Bacterial cell–cell interaction**			
LasR (quorum sensing)	+	+	[64,65]
RhlR (quorum sensing)	+	+	[66,67]

If a virulence factor has been examined and found to facilitate infection in *C. elegans* or in mammals, it is marked with a “+”. If it does not contribute to infection, it is marked with a “−”. If the virulence factor has not yet been studied in the host organism, it is indicated as “n.d.” (not determined).

## Data Availability

No new data were generated in this review.

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
