# Peer review of "Modeling Host–Pathogen Interactions in C. elegans: Lessons Learned from Pseudomonas aeruginosa Infection"

_ijms, 2024, doi:10.3390/ijms25137034_

Round 1

Reviewer 1 Report

Comments and Suggestions for Authors

From 2023 -  5160 publication are listed on term C.elegans Pseudomonas. It in indicate that presented review are dealing with ample of data. By that strictly focus on presented reviewe need to specified. Authors statements: “This review aims to summarize our knowledge on virulence fa tors, infection strategies of Pseudomonas aeruginosa PA14 as well as conserved antimicrobial defenses employed by C. elegans that efficiently respond to pathogen invasion and toxicity.” It indicate that as model only one P.aeruginosa strain will be presented. But is is not the case. Authors do not specified pathogenic factors unique for PA114 strain. Another not fully convincing subjects are comparison mamalien defences systems with C.elegans.  That comparison might good tasks for another manuscript .  In several cases Authors do conclusion that C.elegans lost genes and abilities anti-bacterial potencies , in comparison with mammalin organisms.  Thay are wrong description since evolutionary C. elegans might never have describing function.

In conclusions – Authors need to precisely state out aims or presented review . Presented  data that chosen PA14 strain has unique or common features that convincing readers about value of chosen model. Avoid easy conclusion about similarities of mammalian and  C.elegans anti-bacterial systems.

Author Response

We thank Reviewer #1 for taking the time to review and evaluate our manuscript. We respond to the queries one by one in order of appearance.

We are thankful for the comment regarding the field and focus of our review. Since the landmark first pathogen study by Tan et al (PNAS 1999, 19:715) on Pseudomonas aeruginosa infection of C. elegans 25 years ago, the worm became a versatile infection model. However, there is altogether only 603 scientific articles (90 from 2023) in Pubmed on the term: C elegans Pseudomonas aeruginosa (https://pubmed.ncbi.nlm.nih.gov/?term=C+elegans+Pseudomonas+aeruginosa). Although our manuscript contains over 180 references, it is far from being a systematic review, with a clearly stated focus on PA14 virulence factors and C. elegans pathogen recognition and host defenses. To our knowledge, there is no such work, yet, in the scientific literature. Thus, we believe our work will be of interest to the fields of the C. elegans, innate immunity and Pseudomonas research communities.

We thank for drawing our attention to the misleading statement that only pathogenic factors of the Pseudomonas aeruginosa PA14 strain will be presented. As it is one of the major standard strains used in infection studies both in mammals and in invertebrates, it shares the virulence factors and infection strategies with other Pseudomonas strains, hence the scope of the review is more general. We also corrected the referred sentence by removing ‘PA14’: “This review aims to summarize our knowledge on virulence factors, infection strategies of Pseudomonas aeruginosa“ (line 68).

We thank the comment on the comparison between C. elegans and mammalian defenses against P. aeruginosa. We agree with the Reviewer that a comprehensive comparison, and the description of mammalian immune response per se would be a subject of a separate, longer review beyond the scope of our manuscript. However, we found it important to briefly present the elements of the mammalian innate immune defenses, which could help position C. elegans as a P. aeruginosa infection model and determine how the findings could facilitate mammalian research. Likewise, we aimed to clarify private and shared innate defenses between C. elegans and mammals. To clarify this aim, and also responding to the request of Reviewer 2, we included the key genes of worm defense and whether they have relevance in mammals in new Table 2.

We thank the question regarding our notes on lost genes and pathways by C. elegans. Although these statements sound surprising, we note these are not speculations or conclusions by us, but scientific facts on losing already present immune pathways by nematodes (detailed in a recent review by Pujol and Ewbank, Immunogenetics 2022, 74:63, also cited in the manuscript).

We thank Reviewer #1 again for their work and for the comments. We hope that our reply clarifies the concerns and the revised manuscript will be recommended for publication in IJMS.

Reviewer 2 Report

Comments and Suggestions for Authors

In the manuscript named “Modeling host-pathogen interactions in C. elegans: lessons learned from Pseudomonas aeruginosa infection”, Gábor Hajdú et al have reviewed and summarized genetic regulation processes in response to Pseudomonas aeruginosa infection, including pathogen-associated molecular patterns, surveillance immunity of translation, mitochondria and lysosome-related organelles, involving in regulation of antimicrobial and behavioral defenses by the worm’s neuroendocrine system. Their information would provide valuable cues for molecular mechanism about animal or human response to Pseudomonas aeruginosa infection, and manuscript was also well prepared. However, there were some comments about it.

(1) The key virulence factors were listed in table 1, which was easily accepted by readers, but key genes involving in response to infection process were not listed in table, please summarized them in table, similar to table 1, with refs to each gene.

(2) The organization of some sections was not well, such as section “4 Cell-autonomous stress and immune responses”, in my opinion, it could be displayed with order by “signal infection receive, molecular signal transduction, molecular regulation or function, and organelle reaction”. It would be better in suitable organization.

(3) Similar comments would be suggested about section 5 and 6, “Surveillance immunity of ExoA-mediated translation inhibition”, why the ZIP-2 pathway was detailly described here. The section “6 Surveillance immunity of mitochondrial function”, for example, “responses to induce reactive oxygen species (ROS) generation,” (line 399), which indicated these function genes would be response to ROS, or they were regulated by ROS, the ROS would be upstream in response to infection process. Meanwhile, the ROS would be also effects in infection process, the ROS would be regulated in manuscript.

(4) There were some spell errors, such as: Line 252, “hsf-1” has a formatting error, the subsequent section of “5. Surveillance immunity of ExoA-mediated translation inhibition ”, formatting errors between paragraphs, etc.

(5) References 38 has referencing errors.
